# Learning to represent signals spike by spike

Wieland Brendel[1,2,3‡], Ralph Bourdoukan [2‡], Pietro Vertechi [1,2‡], Christian
K. Machens [1]*, Sophie Denève[2]*

**1** Champalimaud Neuroscience Programme, Champalimaud Foundation, Lisbon, Portugal, **2** Group for
Neural Theory, INSERM U960, Département d'Etudes Cognitives, Ecole Normale Supérieure, Paris, France,
**3** Tübingen AI Center, University of Tübingen, Germany

‡ WB, RB and PV share first authorship on this work. CKM and SD are joint senior authors on this work.
\* christian.machens@neuro.fchampalimaud.org (CKM); sophie.deneve@ens.fr (SD)

## Abstract

Networks based on coordinated spike coding can encode information with high efficiency in
the spike trains of individual neurons. These networks exhibit single-neuron variability and
tuning curves as typically observed in cortex, but paradoxically coincide with a precise, non-
redundant spike-based population code. However, it has remained unclear whether the spe-
cific synaptic connectivities required in these networks can be learnt with local learning
rules. Here, we show how to learn the required architecture. Using coding efficiency as an
objective, we derive spike-timing-dependent learning rules for a recurrent neural network,
and we provide exact solutions for the networks' convergence to an optimal state. As a
result, we deduce an entire network from its input distribution and a firing cost. After learning,
basic biophysical quantities such as voltages, firing thresholds, excitation, inhibition, or
spikes acquire precise functional interpretations.

pcbi.1007692

UNITED STATES

**Data Availability Statement:** The data that support
the findings of this study are openly available in the
GitHub repository at http://github.com/
machenslab/spikes.

## Author summary

Spiking neural networks can encode information with high efficiency in the spike trains
of individual neurons if the synaptic weights between neurons are set to specific, optimal
values. In this regime, the networks exhibit irregular spike trains, high trial-to-trial vari-
ability, and stimulus tuning, as typically observed in cortex. The strong variability on the
level of single neurons paradoxically coincides with a precise, non-redundant, and spike-
based population code. However, it has remained unclear whether the specific synaptic
connectivities required in these spiking networks can be learnt with local learning rules.
In this study, we show how the required architecture can be learnt. We derive local and
biophysically plausible learning rules for recurrent neural networks from first principles.
We show both mathematically and using numerical simulations that these learning rules
drive the networks into the optimal state, and we show that the optimal state is governed
by the statistics of the input signals. After learning, the voltages of individual neurons can
be interpreted as measuring the instantaneous error of the code, given by the error
between the desired output signal and the actual output signal.

**Funding:** This work was funded by the James McDonnell Foundation Award, EU grants BACS FP6-IST- 027140, BIND MECT-CT-20095–024831, and ERC FP7-PREDSPIKE to SD, and the Emmy-Noether grant of the Deutsche Forschungsgemeinschaft (Germany) and a Chaire d'excellence of the Agence National de la Recherche (France) to CKM and an FCT scholarship (PD/BD/105944/2014 Ref.[a] CRM:0022114) to PV. The funders had no role in study design, data collection and analysis, decision to publish, or preparation of the manuscript.

**Competing interests:** The authors have declared that no competing interests exist.

## Introduction

Many neural systems encode information by distributing it across the activities of large populations of spiking neurons. A lot of work has provided pivotal insights into the nature of the resulting population codes [1–4], and their generation through the internal dynamics of neural networks [5–8]. However, it has been much harder to understand how such population codes can emerge in spiking neural networks through learning of synaptic connectivities [9].

For sensory systems, the efficient coding hypothesis has provided a useful guiding principle, which has been successfully applied to the problem of unsupervised learning in feedforward networks [10,11]. When transferring the insights gained in these simplified rate networks to more realistic, biological networks, two key challenges have been encountered. The first challenge comes from locality constraints. Indeed, synapses have usually only access to pre- and postsynaptic information, but most unsupervised learning rules derived in rate networks use omniscient synapses that can pool information from across the network. In turn, the derivation of learning rules under locality constraints has often relied on heuristics or approximations [12–15], although more recent work has shown progress in this area [16–18]. We note that supervised learning in neural networks faces similar problems, and recent work has sought to address these issues [19–23]. We will here focus on unsupervised learning.

The second challenge comes from spikes. Indeed, spikes have often proved quite a nuisance when moving insights from rate networks to spiking networks. In order to maintain the functionality of a given rate network, for instance, the equivalent spiking network usually sacrifices either efficiency or realism. In mean-field approaches, each rate unit is effectively replaced by tens or hundreds of (random) spiking neurons, so that the spiking network becomes a bloated and inefficient approximation of its rate counterpart [24]. In the 'neural engineering framework', this excessive enlargement is avoided [7,25]. However, the spike trains of individual neurons become quite regular, in contrast to the random, almost Poissonian statistics observed in most neural systems.

Some of these problems have recently been addressed in networks with tightly balanced excitation and inhibition [26–29]. These networks can produce functionality with a limited number of neurons and random spiking statistics. One of the key insights of this literature has been that each neuron's voltage should measure a part of the network's global objective, such as the efficiency of the emitted spike code.

However, it has largely remained unclear how networks of spiking neurons could move into this globally optimal regime, given that they are only equipped with local synaptic plasticity rules. We here show that the membrane voltage holds the key to learning the right connectivity under locality constraints. If we start with a randomly connected or unconnected neural network, and simply assume that each neuron's voltage represents part of the global objective, then the locally available quantities such as membrane voltages and excitatory and inhibitory inputs are sufficient to solve the learning problem. Using these ideas, we derive learning rules and prove their convergence to the optimal state. The resulting learning rules are Hebbian and anti-Hebbian spike-timing and voltage-dependent learning rules, and are guaranteed to generate highly efficient spike codes.

## Results

We study a population of excitatory (E) neurons that are interconnected with inhibitory (I) interneurons (Fig 1Ai). The excitatory neurons receive many input signals, $x_j(t)$, from other neurons within the brain, and we will ask how the neurons can learn to encode these signals efficiently in their spiking output. We will first develop a measure for the efficiency of neural population codes, then show the connectivity structure of efficient networks, and then show how the respective connectivity can be learnt. In this work, we focus exclusively on the

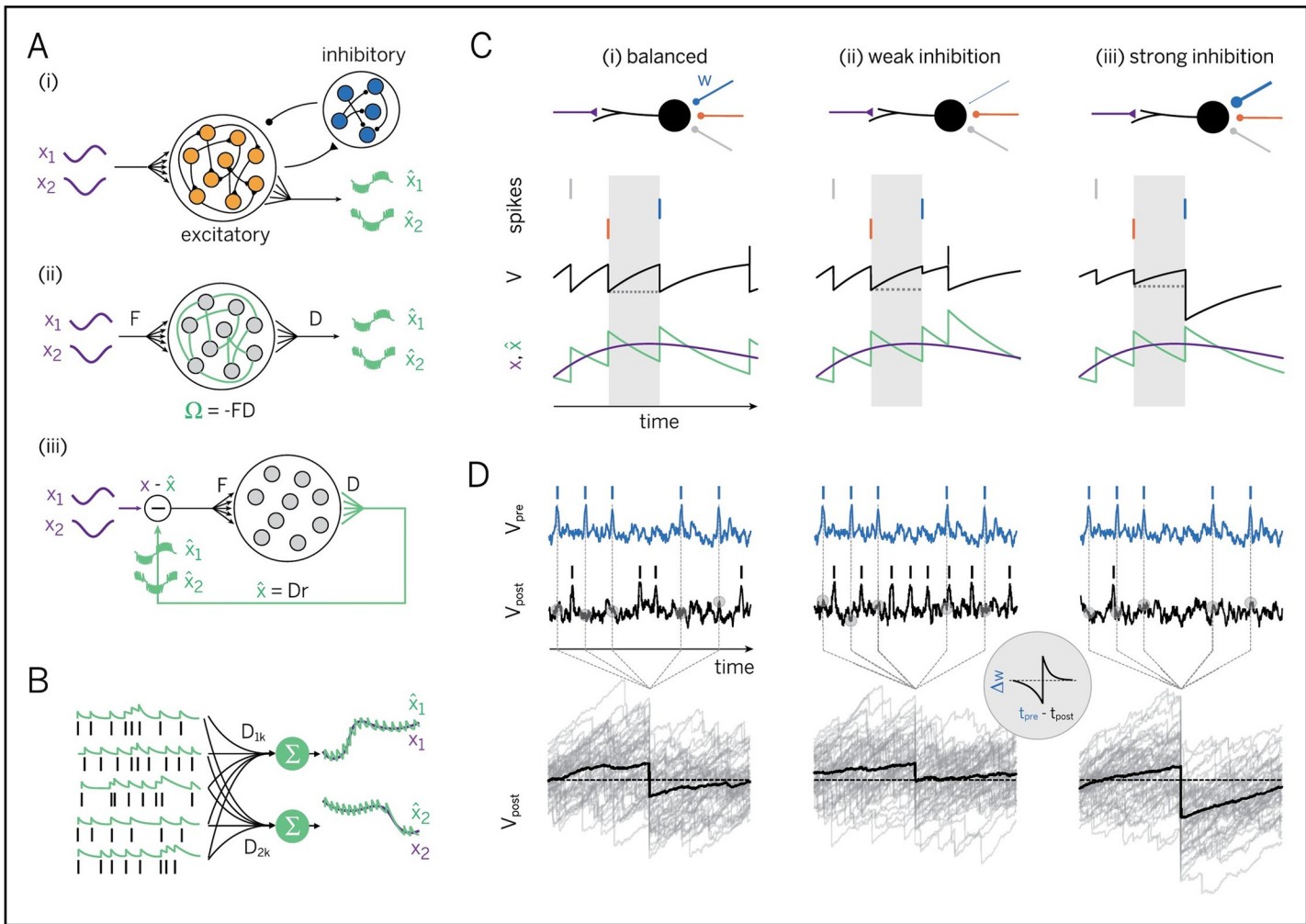

**Fig 1. Learning to represent analog signals efficiently with spikes. A.** (i) Recurrent neural network with input signals $x$ (purple) and signal estimates $\hat{x}$ (green), as reconstructed or read out from the spike spike trains of the excitatory population (see panel B). (ii) Simplified network without separate excitatory and inhibitory populations. ($F$ = matrix of feedforward weights, $D$ = matrix of decoding weights, $\Omega$ = matrix of recurrent weights) A network that has learnt to represent its input signals efficiently should have connectivity $\Omega = -FD$. (iii) Same as (ii), but unfolded to illustrate the effect of the recurrent connections. These connections act to subtract the reconstructed signal estimate from the incoming signal. As a consequence, the net input into each neuron is (a projection of) the reconstruction or coding error, $x - \hat{x}$. **B.** Linear readout of an analog signal from a population of spike trains. The spike train of each neuron is first filtered with a postsynaptic potential (left), and the filtered spike trains are then linearly combined via decoding weights $D$ to yield a signal estimate (right, green traces). In an optimal learnt network, these signal estimates should correspond, as closely as possible, to the input signals (black traces). **C.** Learning of recurrent connections based on precisely balancing the EI currents into each neuron spike by spike. Shown are the neuron's membrane voltage (black), which reflects the coding error, spikes from three inhibitory neurons (vertical lines, color-coded by connection), and the signal (purple), and signal estimate (green). (i) Ideal case with EI balance. Each inhibitory spike perfectly counter-balances the prior excitatory drive. (ii) One inhibitory synapse too weak. The excitatory drive is not perfectly cancelled, resulting in an aberrant, early spike. (iii) One inhibitory synapse too strong. The excitatory drive is over-compensated, resulting in a prolonged hyperpolarization and a delay in subsequent spiking. **D.** Learning of recurrent connections based on minimizing voltage fluctuations. Shown are the voltages and spikes of a pre- and a postsynaptic neuron over a longer time window (top) and the postsynaptic voltage fluctuations aligned at the timing of spikes from the presynaptic neuron (bottom, grey lines), as well as their average (bottom, black line). (i) Ideal case with EI balance. Here, the average effect of the presynaptic spike is to turn a depolarized voltage into an equivalent hyperpolarized voltage (bottom panel, black line). (ii) If the inhibitory synapse is too weak, the average membrane voltage remains depolarized. (iii) If the inhibitory synapse is too strong, the average membrane voltage becomes overly hyperpolarized. (Inset: effect of the derived recurrent plasticity rule when tested with a paired-pulse protocol).

problem of encoding a set of signals, and we defer the problem of how to compute with signals to the discussion.

For concreteness, we will study networks of leaky integrate-and-fire neurons. Each neuron's membrane potential is driven by feedforward input signals, $x_j(t)$, which we will model as a leaky integral of input currents $c_j(t)$, and by recurrent inputs that feed the output spike trains,

$o_k(t)$, back into the network. For simplicity, we will ignore the inhibitory interneurons for now and treat them as simple relays (Fig 1Aii). As a consequence, we allow the excitatory neurons to violate Dale's law, a problem we will come back to later. Formally, the membrane voltages of the excitatory neurons obey the equation

$$\frac{dV_i}{dt} = -V_i + \sum_{j=1}^{M} F_{ij} c_j(t) + \sum_{k=1}^{N} \Omega_{ik} o_k(t),$$

(Eq 1)

where $F_{ij}$ are the feedforward weights, and $\Omega_{ik}$ contains the recurrent synapses (for $i \neq k$) and the voltage resets (for $i = k$). A spike is fired when the voltage surpasses a threshold, $T_i$. The voltage is then reset to the value $V_i = T_i + \Omega_{ii}$, and we assume that $\Omega_{ii} < 0$. For simplicity, here we consider *instantaneous* synaptic transmission: the impact of synaptic delays on the network will be examined in Fig 7.

The first objective of the network will be to encode the input signals into a spiking output such that a downstream observer can reconstruct the input signal through a linear readout, i.e., a weighted sum of the neural responses (Fig 1B). We define this linear readout as

$$\hat{x}_j(t) = \sum_{k=1}^{N} D_{jk} r_k(t),$$

(Eq 2)

where $r_k(t)$ is the postsynaptically filtered spike train of the $k$-th excitatory neuron, and $D_{jk}$ is the decoding weight associated with the $j$-th signal.

The second objective of the network will be to find, among all possible spiking outputs, and all possible decoders, the ones that are the most efficient. We define the coding efficiency of the population as a trade-off between the accuracy and the cost of the generated code,

$$E = \langle \sum_{j=1}^{M} (x_j - \hat{x}_j)^2 + C(r) \rangle,$$

(Eq 3)

where the angular brackets denote averaging over time. The first term measures the accuracy of the code, given by the mean-squared error between the input signals and the linear readout. The second term, $C(r)$, denotes the cost of the code, exemplified for instance by the number of spikes fired. The smaller the loss, the higher the coding efficiency (see S1 Text, Section1, for details).

## Efficient spike coding requires balance of excitation and inhibition

To find the most efficient spiking output, our network will need to modify its synapses. Since a single synapse can only see its pre- and postsynaptic partners and their relative spike trains, it cannot perceive the coding efficiency of the whole network. Without that information, it is unclear how the synapse should modify its weights in order to improve the coding efficiency. This rift between locally available information and global objective is the key conundrum of synaptic plasticity.

However, imagine we could intervene and simply set each neuron's recurrent synaptic weights such that they become equal to the feedforward weights multiplied by the decoding weights of a downstream observer, i.e., $\Omega_{ik} = -\sum_j F_{ij} D_{jk}$. As shown in S1 Text, Sections 2 and 3, the membrane potential of each neuron can then be rewritten as

$$V_i(t) = \sum_{j=1}^{N} F_{ij}(x_j(t) - \hat{x}_j(t)).$$

(Eq 4)

In other words, given this specific connectivity structure, each neuron's membrane potential suddenly reflects a component of the global coding error, given by the difference between the input signals, $x_j(t)$, and the linear readout of a hypothetical downstream area, $\hat{x}_j(t)$. This peculiar structure emerges even though the membrane potential is generated from only feedforward and recurrent inputs (Fig 1Aii and 1Aiii). Since synaptic plasticity can sense postsynaptic voltages, synapses have gained unexpected access to a component of the global coding error.

Moreover, each neuron will now bound its component of the error from above. Each time the error component becomes too large, e.g., due to an excitatory signal input, the membrane potential reaches threshold, and the neuron fires. The spike changes the readout, and the global coding error decreases (under reasonable conditions on $F_{ij}$ and $D_{jk}$, see S1 Text, Section 4). This decrease in error is then signaled throughout the network. First, the firing neuron resets its own voltage after the spike, thus signaling to itself that its error component has decreased. Second, the firing neuron inhibits (or excites) all neurons with similar (or opposite) feedforward inputs, thus signaling them the decrease in error. The concurrent change in their respective membrane voltages is proportional to the overlap in information and thereby reflects the required update of the error components they are responsible for.

As a consequence, excitatory inputs that depolarize the membrane potential signal growing coding errors. Vice versa, inhibitory inputs that repolarize the membrane potential signal shrinking coding errors. In turn, when coding errors are kept in check, each feedforward excitatory input will be counterbalanced by a recurrent inhibitory input (and vice versa). This latter reasoning links the precision of each neuron's code to the known condition of excitatory and inhibitory (EI) balance [26,30–33]. Indeed, if excitatory and inhibitory inputs are balanced optimally, the variance of the membrane potential, and thus, each neuron's error component, is minimized.

## Recurrent synapses learn to balance a neuron's inputs

How can a network of neurons learn to move into this very specific regime? Several learning rules for EI balance have been successfully proposed before [34,35], and spike-timing-dependent plasticity (STDP) can even balance EI currents on a short time scale [35]. We will show that learning to balance excitatory and inhibitory inputs does indeed lead to the right type of connectivity (Fig 1Aii and 1Aiii), as long as EI currents are balanced as precisely as possible. Learning to balance avoids the pitfalls of a direct optimization of the coding efficiency with respect to the decoder weights, which is mathematically possible, but biophysically implausible for the synapses we consider here (see S1 Text, Section 5). We developed two ways of reaching the balanced regime (see S1 Text, Section 6 for a high-level, technical overview). The first scheme balances excitatory and inhibitory currents on a fine time scale (see S1 Text, Sections 7 and 8 for details), while the second scheme minimizes the voltage fluctuations (see S1 Text, Sections 9–12 for details). We here briefly explain the current-based scheme, but then focus on the voltage-based scheme for the rest of the text.

The first scheme directly targets the balance of excitatory and inhibitory currents. In Fig 1C, we show a neuron that receives excitatory feedforward inputs and inhibitory recurrent inputs. In the interval between two inhibitory spikes, the neuron integrates its excitatory feedforward input currents, which leads to a transfer of electric charges across the membrane (Fig 1Ci, gray area). When the next inhibitory spike arrives (Fig 1Ci, blue), electric charges are transferred in the opposite direction. Precise EI balance is given when these two charge transfers cancel exactly. When the second inhibitory spike overshoots (undershoots) its target, then the respective synaptic weight was too strong (weak), see Fig 1Cii and 1Ciii. To reach precise

EI balance, this weight therefore needs to be weakened (strengthened). This learning scheme keeps the neuron's voltage (and thereby its component of the coding error) perfectly in check (see S1 Text, Sections 7 and 8 for details). We note that the membrane potential shown in Fig 1C is an illustrative toy example, for a network representing only one input signal with four neurons. In larger networks that represent several input signals, the membrane potentials become more complex, and the inhibitory inputs due to recurrent connections become weaker than the voltage reset after a spike (see also below).

The precise accounting of charge balances across the membrane may seem unfeasible for real neurons. Our second scheme minimizes charge imbalances by confining deviations from a neuron's resting potential. If a recurrent weight is set such that each presynaptic spike, on average, resets a voltage depolarization to an equivalent hyperpolarization (or vice versa), then the membrane voltage is maximally confined (see Fig 1D). To move a recurrent synapse into this state, its weight should be updated each time a spike from presynaptic neuron $k$ arrives, so that

$$\frac{d\Omega_{ik}}{dt} \propto \text{pre} \times \text{post} = -o_k(2V_i + \Omega_{ik}). \tag{Eq 5}$$

where $o_k$ is the presynaptic spike train and $V_i$ is the postsynaptic membrane potential before the arrival of the presynaptic spike. According to this rule, the recurrent connections are updated only at the time of a presynaptic spike, and their weights are increased and decreased depending on the resulting postsynaptic voltage. While this rule was derived from first principles, we note that its multiplication of presynaptic spikes and postsynaptic voltages is exactly what was proposed as a canonical plasticity rule for STDP from a biophysical perspective [36]. A minor difference to this biophysically realistic, 'bottom-up' rule, is that our rule treats LTP and LTD under a single umbrella. Furthermore, our rule does not impose a threshold on learning.

Once a synapse has been learnt with this voltage-based learning rule, it will tightly confine all voltage fluctuations as much as possible. This average confinement is illustrated in Fig 1D. We note that the membrane potentials look more realistic here simply because the illustration is based on the simulation of a larger network with multiple input signals.

The learning rule drives the recurrent weights to the desired connectivity, given by the multiplication of the feedforward weights, $F_{ij}$, with an (a priori unknown) decoder matrix, $D_{jk}$, see Fig 1Aii and 1Aiii. To gain some intuition as to why that is the case, we will show that this connectivity structure is a stationary point of the learning rule. At this stationary point, the recurrent weights are no longer updated and become proportional to the average postsynaptic voltage of neuron $i$, $\Omega_{ik} = -2\langle V_i \rangle_k$, where the average, denoted by the angular brackets, is taken over all time points directly before the arrival of a spike from the presynaptic neuron $k$ (see Fig 1Di). Since, whenever $\Omega_{ik} = -\sum_j F_{ij} D_{jk}$, the connectivity structure dictates that the voltage becomes a function of the global coding error, as stated in Eq 4, the stationary point can be rewritten as $\Omega_{ik} = -2\sum_j F_{ij} \langle x_j - \hat{x}_j \rangle_k$. If we now simply define the decoder matrix as $D_{jk} = 2\langle x_j - \hat{x}_j \rangle_k$, then $\Omega_{ik} = -\sum_j F_{ij} D_{jk}$. Accordingly, the peculiar multiplicative form of the recurrent weights, which transformed the voltage into a component of the coding error, is a stationary point of the learning rule (see S1 Text, Section 9 for details and an additional convergence proof).

Depending on the precise cost terms, $C(r)$, required by the loss function, the learning rules undergo slight modifications. The effect of these cost terms is to penalize both the total number of spikes fired by the network, as well as high firing rates in individual cells. The learning

rules used in all simulations are of the form

$$\frac{d\Omega_{ik}}{dt} \propto -o_k(\beta(V_i + \mu r_i) + \Omega_{ik} + \mu \delta_{ik}). \qquad \text{(Eq 6)}$$

with $\beta$ and $\mu$ positive constants, and with $\delta_{ik}$ the Kronecker delta (see S1 Text, Sections 9–13 for a detailed explanation of these modifications and their relation to the cost).

Fig 2 illustrates the effect of the voltage-based learning rule in a network with 20 neurons receiving two random, uncorrelated feedforward inputs (see S1 Text, Section 14 for details on the simulations). Since each neuron receives two input signals, each neuron has two feedforward weights. The initial setting of these weights was lopsided, as shown in Fig 2Bi (left panel), so that no neuron received a positive contribution of the first input signal. The recurrent weights were initially set equal to zero (Fig 2Bi, right panel; the diagonal elements correspond to the self-resets of the neurons).

While the network receives the random input signals, the recurrent synapses change according to the learning rule, Eq 6, and each neuron thereby learns to balance its input currents. Once learnt, the recurrent connectivity reaches the desired structure, and the voltages of the neurons become proportional to a component of the coding error. As a result of the EI balance, the voltage fluctuations of individual neurons are much better bounded around the resting potential (compare Fig 2Ei with 2Eii), the global coding error decreases (Fig 2A), and the network experiences a large drop in the overall firing rates (Fig 2A, 2Di and 2Dii). The network's coding improvement is best illustrated in Fig 2Ci and 2Cii, where we test the network with two input signals, a sine and cosine, and illustrate both the input signals and their reconstructions, as retrieved from the spike trains in Fig 2Di and 2Dii using an optimal decoder. Note that this improvement occurred despite a drastic drop in overall firing rates (Fig 2Di and 2Dii).

## Feedforward weights change to strengthen postsynaptic firing

Despite the performance increase, however, the network still fails to represent part of the input, even after the recurrent connections have been learnt (Fig 2Cii, arrow). Indeed, in the example provided, positive values of the first signal cannot be represented, because there are no neurons with positive feedforward weights for the first signal (Fig 2Bi and 2Bii). These missing neurons can be easily spotted when plotting the tuning curves of all neurons (Fig 2Gi and 2Gii). Here, directions of the input signal associated with positive values of the first signal are not properly covered, even after the recurrent weights have been learnt (Fig 2Gii, arrow).

Consequently, the feedforward connections need to change as well, so that all parts of the input space are dealt with. We can again obtain a crucial insight by considering the final, 'learnt' state, in which case the feedforward connections are directly related to the optimal decoding weights. For example, if the input signals are mutually uncorrelated, i.e., $\langle x_i(t)x_j(t)\rangle = 0$ for zero-mean inputs and $i \neq j$, then the optimal feed-forward and decoding weights are equal, i.e., $F_{ik} = D_{ki}$ (see S1 Text, Section 4). In Fig 3A, we illustrate the problem with five neurons that seek to represent two input signals. We assume a constant input signal, which we represent by a point in a signal space (Fig 3Ai, purple dot). In turn, a neuron's spiking shifts the signal estimate in a direction given by its respective decoding weights, which we can illustrate through vectors (Fig 3Ai, colored arrows). Accordingly, the input signal can be represented by a linear combination of the decoding vectors. For a biased distribution of decoding vectors, some input signals will require the combined effort of many neurons (Fig 3Ai). For uncorrelated input signals, however, the best representation is achieved when the decoding vectors (and thereby the feedforward weights) are evenly distributed (Fig 3Aii).

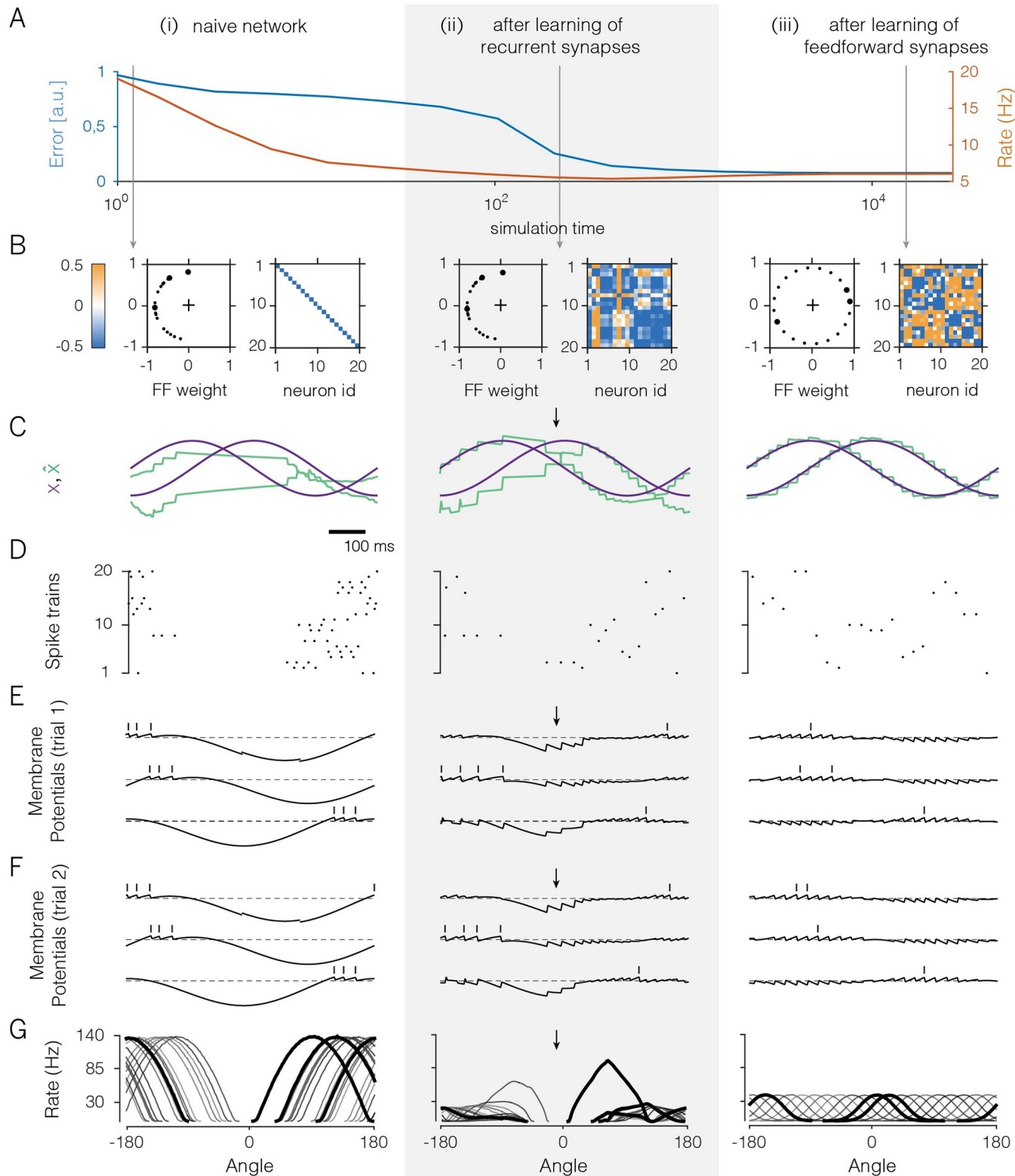

**Fig 2. A 20-neuron network that learns to encode two randomly varying signals. A.** Evolution of coding error (blue), defined as the mean-square error between input signal and signal estimate, and mean population firing rate (orange) over learning. **B.** Feedforward and recurrent connectivity at three stages of learning. In each column, the left panel shows the two feedforward weights of each neuron as a dot in a two-dimensional space, and the right panel shows the matrix of recurrent weights. Here, off-diagonal elements correspond to synaptic weights (initially set to zero), and diagonal elements correspond to the

neurons' self-resets after a spike (initially set to -0.5). **C.** Time-varying test input signals (purple) and signal estimates (green). The test signals are a sine wave and a cosine wave. Signal estimates in the naive network are constructed using an optimal linear decoder. Arrows indicate parts of the signal space that remain poorly represented, even after learning of the recurrent weights. **D.** Spike rasters from the network. **E.** Voltages and spike times of three exemplary neurons (see thick dots in panel B). Dashed lines illustrate the resting potential. Over the course of learning, voltage traces become confined around the resting potential. **F.** As in **E**, but for a different trial. **G.** Tuning curves (firing rates as a function of the angle of an input signal with constant radius in polar coordinates for all neurons in the network. Angles from $-90°$ to $90°$ correspond to positive values of $x_1$ which are initially not represented (panel B).

The feedforward weights of the $i$-th neuron can learn to optimally cover the input space if they change each time neuron $i$ fires a spike,

$$\frac{dF_{ij}}{dt} \propto \left( x_j - \alpha F_{ij} \right) o_i, \tag{Eq 7}$$

where $x_j$ is the feed-forward input signal, $\alpha$ is a positive constant whose value depends on the enforced cost (see S1 Text, Section 11), and $o_i$ is the neuron's spike train. Note that the feedforward weights remain unchanged if neuron $i$ does not spike.

The intuition for this rule is shown in Fig 3B. In an unconnected network, a neuron fires the most when its feedforward input drive is maximal. Under a power constraint on the input signal, the drive is maximized when the vector of input signals aligns with the vector of feedforward weights. In a network connected through recurrent inhibition, however, neurons start competing with each other, and a neuron's maximum firing (Fig 3Bi; dashed lines) can shift away from the maximum input drive (Fig 3Bi, colored arrows) towards stimuli that face less competition. If competition is well-balanced, on the other hand, then a neuron's maximum firing will align with the maximum input drive, despite the presence of recurrent connections (Fig 3Bii, compare colored arrows and dashed lines). The above learning rule moves the network into this regime by shifting the feedforward weights towards input signals that elicit the most postsynaptic spikes (Fig 3Bii, gray arrows). Learning converges when all tuning curve maxima are aligned with the respective feedforward weights (Fig 3Bii; dashed lines and arrows). Eventually, the input space is thereby evenly covered (see S1 Text, Section 10 for mathematical details).

From the perspective of standard frequency-modulated plasticity, the learning rule is Hebbian: whenever neuron $i$ fires a spike, the resulting change in its synaptic weight $F_{ij}$ is proportional to the $j$-th presynaptic input, $x_j$, received at that time. The more neuron $i$ spikes, and the higher the input $x_j$, the stronger the change in weight. Accordingly, connections are reinforced for co-occurring high pre- and postsynaptic activity. In the case of correlated input signals, the term "$F_{ij}$" is replaced by the covariance of the $j$-th presynaptic input signal with the total postsynaptic input current (see S1 Text, Section 12). In this case, the decoding weights provide optimal coverage by favoring more frequent input signal directions (See Fig 3C and 3D).

The effect of the feedforward plasticity rule is shown in Fig 2Aiii–2Giii. The feedforward weights change slowly until the input space is spanned more uniformly (Fig 2Biii). While these changes are occurring, the recurrent weights remain plastic on a faster time scale and thereby keep the system in a balanced state. At the end of learning, the neuron's tuning curves are uniformally distributed (Fig 2Giii), and the quality of the representation becomes optimal for all input signals (Fig 2Aiii and 2Ciii). More specifically, the feedforward weights have become identical to the decoding weights, $F_{ik} = D_{ki}$, and the latter minimize the objective function, Eq 3.

Importantly, the final population code represents the input signals spike by spike, with a precision that approaches the discretization limit imposed by the spikes, i.e., the unavoidable steps in the signal estimate caused by the firing of individual spikes. Initially, when the neurons were unconnected (Fig 2Bi), their voltages reflected the smooth, time-varying input (Fig 2Ei).

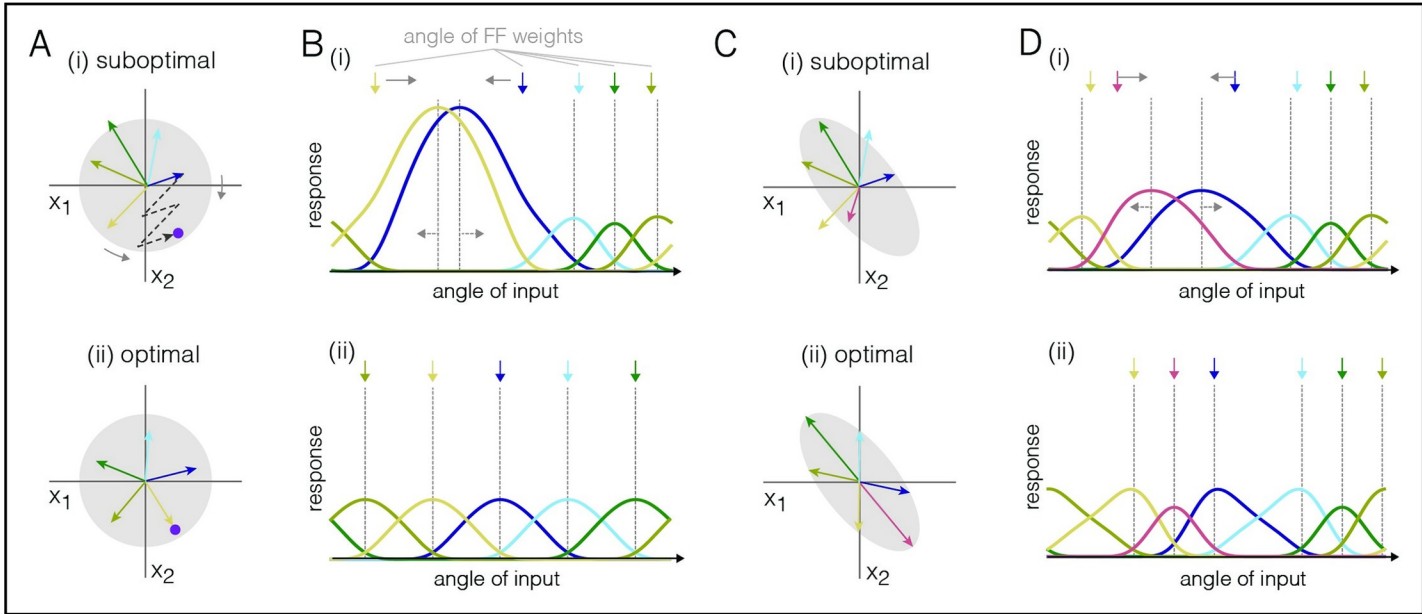

**Fig 3. Learning rules for the feedforward weights, depicted for a network with five neurons. A.** Arrangement of decoding weights influences coding efficiency. The purple dot represents the input signals, and each vector represents the jump in the signal estimates caused by the firing of one neuron. The gray circle represents the distribution of input signals; here, they are centered and uncorrelated. (i) A biased arrangement of the decoding weights is suboptimal for uncorrelated signals. Many spikes are required to represent the purple input. (ii) Evenly spaced decoding weights are optimal for uncorrelated signals. Here, the purple input can be reached with a single spike. **B.** Tuning curves of the five neurons before and after training. Shown are the firing rates of the neurons as a function of the angle of the input signal. Colored arrows above represent the feedforward weights (or the input signals that drive the neurons maximally in the absence of recurrent connections). (i) In the untrained network, maximum input drive and maximum firing are not aligned. The learning rule shifts the feedforward weights towards the maximum of the firing rates (gray arrows, top). In turn, the firing rate maxima shift in the opposite direction (gray arrows, bottom). (ii) After learning, the maximum input drive (and thereby the feedforward weights) are aligned with the maximum firing rate. **C.** Similar to **A**, but for correlated input signals. **D.** Similar to **C**, but for correlated input signals. In the optimal scenario, the neurons' feedforward and decoding weights are attracted towards more frequent stimuli.

Moreover, neurons fired their spikes at roughly the same time from trial to trial (compare Fig 2Ei with 2Eii). After learning, the membrane potentials are correlated, reflecting their shared inputs, yet the individual spikes are far more susceptible to random fluctuations (compare Fig 2Eiii with 2Fiii). Indeed, whichever neuron happens to fire first immediately inhibits (resets) the others, so that a small initial difference in the membrane potentials is sufficient to change the firing order completely. Here, the random nature of spike timing is simply a consequence of a mechanism that prevents any redundant (or synchronous) spikes. More generally, any source of noise or dependency on previous spike history will change the firing order, but without a significant impact on the precision of the code. Thus, variable spike trains co-exist with a highly reproducible and precise population code.

## Learning in networks with separate excitatory and inhibitory populations

We have so far ignored Dale's law so that individual neurons could both excite and inhibit other neurons. Fortunately, all of our results so far can also be obtained in networks with separate excitatory (E) and inhibitory (I) populations (Fig 1Ai), governed by Eq 1. In this more realistic case, the inhibitory population must simply learn to represent the population response of the excitatory population, after which it can balance the excitatory population in turn. This can be achieved if we train the EI connections using the feedforward rule (Eq 7) while the II, EE, and IE connections are trained using the recurrent rule (Eq 6; see S1 Text, Section 13 for details).

Fig 4 illustrates how the key results obtained in Fig 2 hold in the full EI network. The network converges to the optimal balanced state (Fig 4B), and the precision of the representation improves substantially and approaches the discretization limit (Fig 4Bi and 4Cii), despite the overall decrease in output firing rates (Fig 4Bii and 4Cii). Initially regular and reproducible spike trains (Fig 4Biii) become asynchronous, irregular, and comparable to independent Poisson processes (Fig 4Biii, pairwise correlations are smaller than 0.001). Crucially, both the inhibitory and excitatory populations provide an accurate representation of their respective input signals, as shown by their small coding errors (Fig 4Bi). Furthermore, we observe that the neurons' tuning curves, when measured along the first two signal directions, are bell-shaped just as in the previous example (Fig 4Dii). Note that the inhibitory neurons fire more and have broader tuning than the excitatory neurons. This result is simply owed to their smaller number: since less neurons are available to span the signal space with their feedforward weights, they generally face less competition, and consequently have broader tuning.

## Learning for correlated inputs

We have so far considered input signals that are mutually uncorrelated. For correlated input signals, the network learns to align its feedforward weights to the more frequent signal directions (Fig 3C). As a result, the tuning curves of the learnt network reflect the distribution of inputs experienced by the network (Fig 3D). In particular, tuning curves are denser and sharper for signal directions that are a-priori more probable. This result is reminiscent of the predictions for efficient rate-based population codes with independent Poisson noise [37]. Note, however, that our networks learn a spike-per-spike code that is more precise and efficient than rate-based population codes.

To further demonstrate the power of the learning rules, using learning rules developed in S1 Text, Section 12, we trained a network to represent speech signals, filtered through 25 frequency channels, in its spiking output (Fig 5A). Despite consisting of 100 neurons that fire at only ~ 4 Hz, the network learns to represent the signals with high precision (Fig 5B and 5C). This feat would be impossible if the network had not learnt the strong correlations in speech. After training with the speech signals, the feedforward and decoding weights adopt a structure reflecting the natural statistics of speech. The feedforward weights typically have excitatory subfields covering a limited range of frequencies, as well as inhibitory subfields (Fig 6Ai and 6Aii). Decoding weights are wider and more complex, thus exploiting the high correlations between frequency channels (Fig 6Bi and 6Bii). These model predictions are broadly compatible with observations in the mammalian auditory pathway, and notably the representation of speech signals in A1 [38].

As a drawback, the network has become specialized, and a new "non-speech" stimulus results in poor EI balance, high firing rates, and poor coding (Fig 5D and 5E). After experiencing the new sound several times, however, the network represents the "non-speech" sound as precisely and parsimoniously as the previously experienced speech sounds (Fig 5F). After retraining to the new stimulus, feed-forward weights are modified specifically at the frequencies of the new stimulus (Fig 6Aiii). However, these changes are not massive. In particular, only a handful of neurons (two in this example) have become truly specialized to the new stimulus, as reflected by their decoding weights (Fig 6Biii).

## Robustness of Learning against perturbations

A crucial final question is whether these learning rules continue to work under more realistic conditions, such as noise in various components of the circuit, delays in the synaptic transmission, or constraints on the ability of arbitrary neurons to form synaptic connections in the first

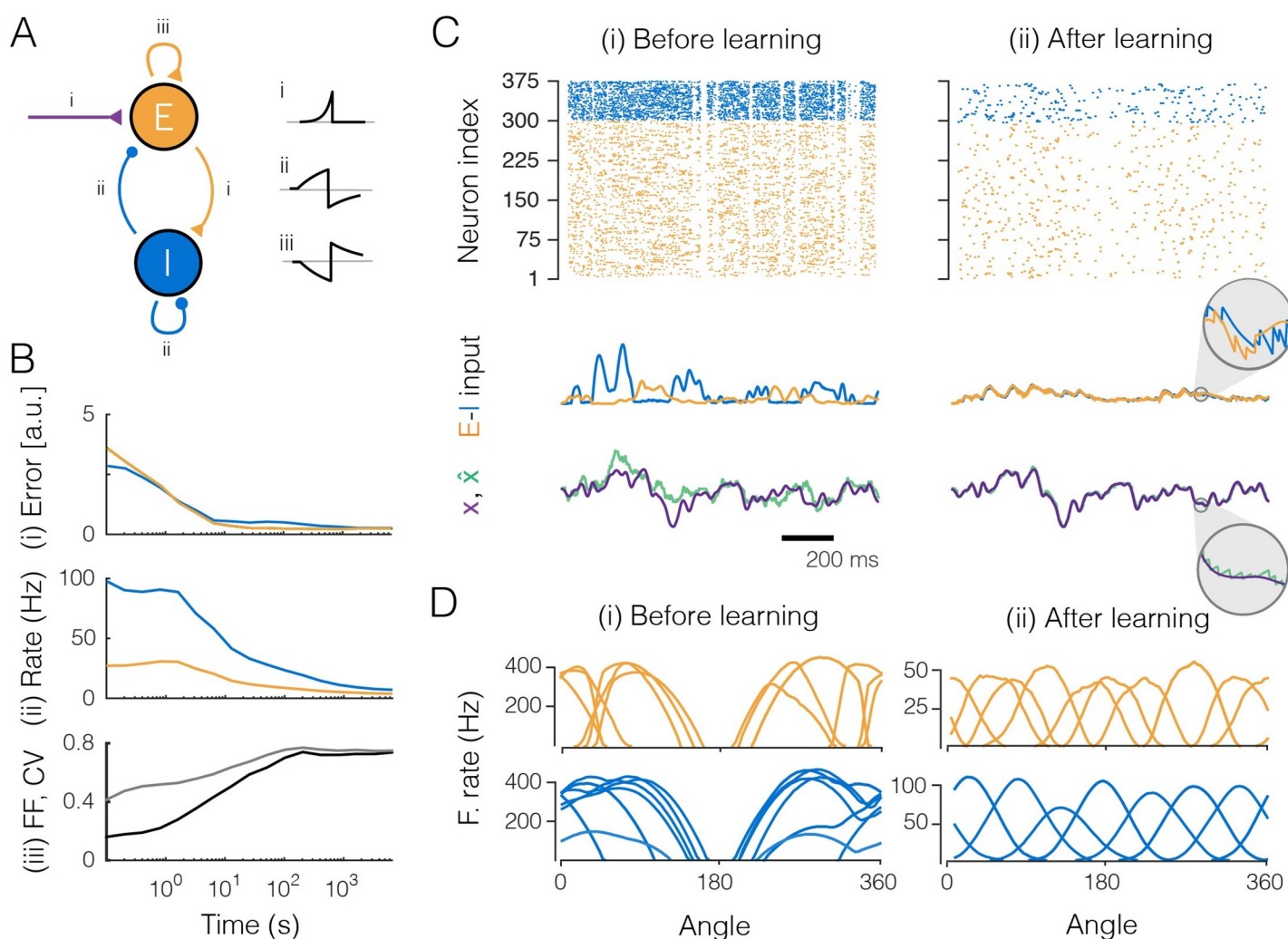

**Fig 4. Large network (300 excitatory and 75 inhibitory neurons) that learns to encode three input signals.** Excitation shown in orange, inhibition in blue. **A.** The EI network as in Fig 1Ai and the learning rules (i, feedforward rule; ii and iii, recurrent rule). The insets show cartoon illustrations of the learning rules, stemming from STDP-like protocols between pairs of neurons, with the x-axis representing the relative timing between pre- and postsynaptic spikes ($\Delta t = t_{pre} - t_{post}$), and the y-axis the change in (absolute) weight. Note that increases (decreases) in synaptic weights in the learning rules map onto increases (decreases) for excitatory weights and decreases (increases) for inhibitory weights. This sign flip explains why the STDP-like protocol for EE connections yields a mirrored curve. **B.** Evolution of the network during learning. (i) Coding error for excitatory and inhibitory populations. The coding error is here computed as the mean square error between the input signals and the signal estimates, as reconstructed from the spike trains of either the excitatory or the inhibitory population. (ii) Mean firing rate of excitatory and inhibitory populations. (iii) Averaged coefficient of variation (CV, gray) and Fano factor (FF, black) of the spike trains. **C.** Network input and output before (i) and after (ii) learning. (Top) Raster plots of spike trains from excitatory and inhibitory populations. (Center) Excitatory and inhibitory currents into one example neuron. After learning, inhibitory currents tightly balance excitatory currents (inset). (Bottom) One of the three input signals (purple) and the corresponding signal estimate (green) from the excitatory population. **D.** Tuning curves (firing rates as a function of the angle for two of the input signals, with the third signal clamped to zero) of the most active excitatory and inhibitory neurons.

place. To answer these questions, we first note that the learning rules work independent of the initial state of the network. As long as the initial network dynamics are sufficiently stable, the learning rules converge globally (see S1 Text, Sections 9 and 10 for a proof). We furthermore note that the networks perform better and become more robust as the number of neurons increases (Fig 7A, see also [28]).

We first studied how the learning rules perform when not all neurons can form (potential) synaptic connections. As shown in Fig 7B, eliminating potential synapses only affects the performance of the learnt network when drastic limits are imposed (less than 20% of connections

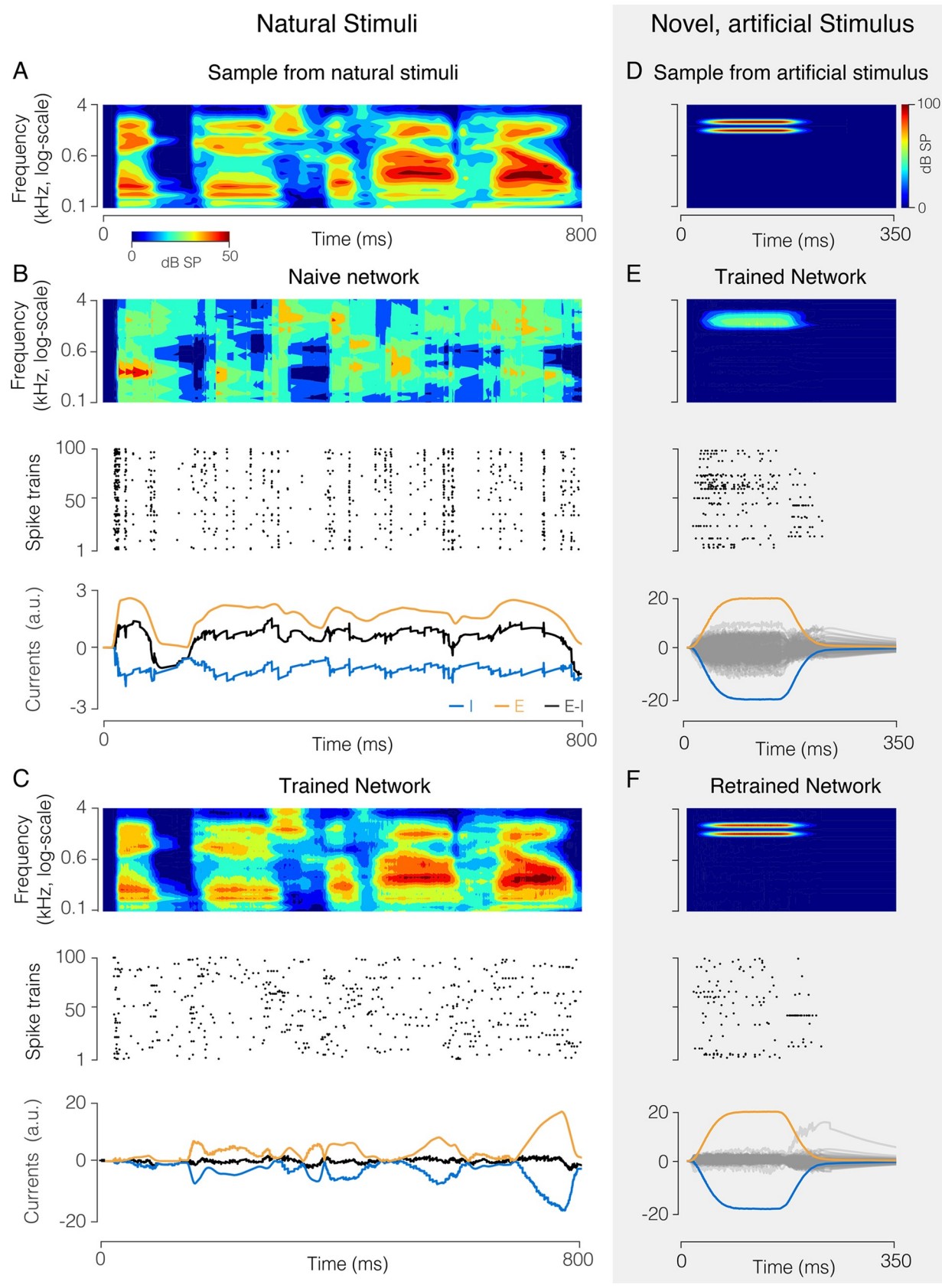

**Fig 5. Network (100 neurons) that encodes a high-dimensional, structured natural input (speech sounds). A.** Spectrogram of a speech sound. **B.** "Naive" network with random feedforward and recurrent weights. (Top) Optimal linear estimator applied to output spike trains reconstructs the stimulus poorly. (Center) Spike raster from all neurons, showing synchronous firing. (Bottom panel) Excitatory (orange) and inhibitory (blue) current into an example neuron are poorly balanced, causing large fluctuations in the total current (black). **C.** Same as **B**, after learning. The signal estimate tracks the signal closely (top), spike trains are asynchronous and irregular (center), and EI currents are tightly balanced (bottom). **D.** Spectrogram of artificial, "non-speech" sound. **E.** Response of the trained network trained to a non-speech sound, similar format as **B**, **C**. The new sound is improperly reconstructed (top), and EI currents of individual neurons are poorly balanced (bottom). Grey lines show the total (E+I) currents for the individual neurons, orange and blue lines show the mean excitatory and inhibitory currents, averaged over the population. **F.** Same as **E**, after re-training the network with a mixture of speech sounds and the new sound. The new sound is now represented precisely (top) with fewer spikes (center), and EI balance is improved (bottom).

available for a network with $N = 50$ inhibitory neurons). Smaller networks are generally more sensitive (Fig 7B, dashed blue line), whereas larger networks are less sensitive.

To study the resistance of the learning rules against noise, we introduced random currents into the neurons, which can be viewed as a simulation of stochastic fluctuations in ion channels or background synaptic activity. For reasonable levels of noise, this modification had an essentially negligible effect on network performance. Fig 7C shows the error made by the networks after learning as a function of the strength of the introduced noise.

A final concern could be to what extent the learning rules rely on overtly simplistic synaptic dynamics—each spike causes a jump in the postsynaptic voltage followed by an exponential decay. To address this question, we also simulated the network assuming more realistic synaptic dynamics (Fig 7D). We measures the effective delay of transmission as the time-to-peak for a postsynaptic potential. Within the range of mono-synaptic transmission delays observed in cortical microcircuits, the networks still learns to encode their input signals efficiently, see Fig 7B. As transmission delays grow larger, a degradation in performance is incurred due to limited synchrony between similarly tuned neurons, which is unavoidable in the presence of delayed inhibition. Indeed, by keeping excitatory and inhibitory currents as balanced as possible, the network automatically finds an optimal regime of weak synchronization, removing the need for fine tuning of the network parameters. Such weak synchronization causes weak oscillations in the network activity whose time scales may be related to gamma rhythms [29].

Thus, we found that under a wide range of perturbations, the network learnt to achieve a performance near the discretization limit, outperforming conventional spiking networks or population coding models based on Poisson spike trains. This robustness is inherited from the generality of the relationship between EI balance and the error-correcting coding strategy in the network [27,28].

## Manipulating plasticity

One of the key consequences of our derivations is that feedforward and recurrent plasticity serve different goals. Whereas recurrent plasticity works to balance the network, keeping all voltages (and thereby the respective coding errors) in check, feedforward plasticity works to unbalance each neuron, driving up excitation as much as possible. Since the recurrent plasticity rules are faster, they win this competition, and the network remains in a balanced state.

These considerations lead to some fundamental, yet experimentally testable predictions that are illustrated in Fig 8. In this simulated experiment, a number of neurons with similar tuning curves to angular stimuli (such as oriented gratings) are suddenly killed (Fig 8A, dashed arrows). In principle, this should severely impair the representation of stimuli in this direction. However, three mechanisms are recruited to compensate for the degradation of the representation. In a first step, the EI imbalance introduced by the lesion is immediately corrected by the network. This occurs instantaneously, before any plasticity mechanisms can be involved. As a result, the tuning curves of some neurons shift, widen, and increase in amplitude in an

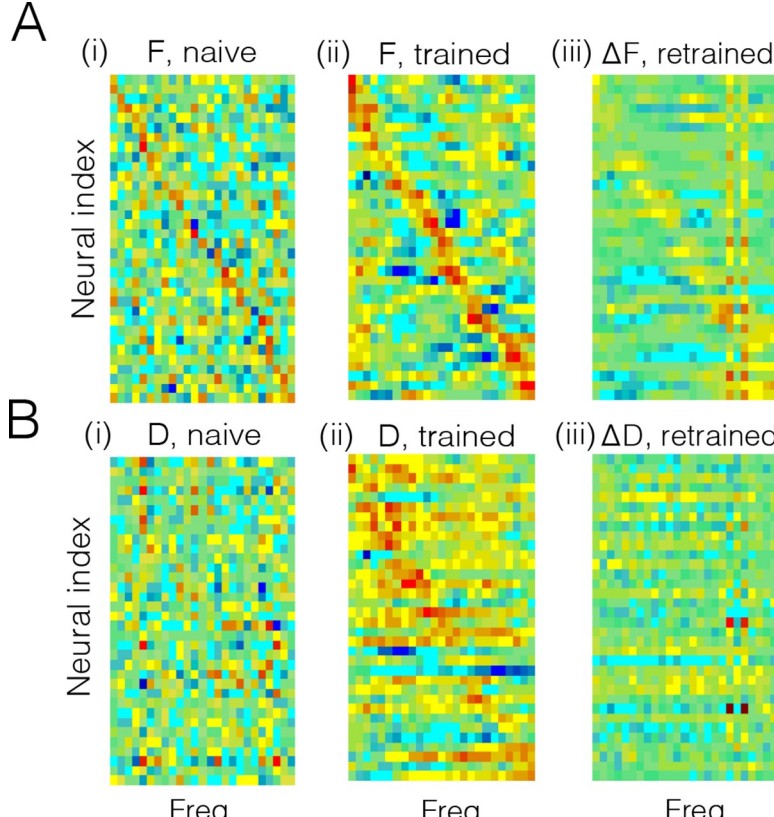

**Fig 6. Feedforward and recurrent connection structure before and after learning speech sounds. A.** (i) Feedforward weights of neurons before learning. These weights correspond to the spectral receptive fields (SRF) of the neurons, since they weight the different frequency bands. Although set to random, a weak diagonal is visible because neurons were sorted according to maximal frequency. Bluish colors correspond to negative values, reddish colors to positive values. (ii) Feedforward weights or SRFs after learning. The SRFs now have an excitatory subfield, and one or two inhibitory subfields, compatible with SRFs observed in primary auditory cortex [38]. Note that the neurons have been resorted according to maximal frequency. (iii) Change in SRFs after re-training with a new stimulus (see Fig 5D). The SRFs change selectively (positively and negatively) at the position of the trained frequencies. The frequency-selective change in SRFs is in line with fast plastic changes of SRFs observed following behavioral training [39]. There is also a small decrease in gain at other frequencies, due to the competition with the new stimulus. **B.** Same as in A, but for the decoding weights. (i) Decoding weights before learning appear random. Here, we sorted neurons as in A(i) in order to ease the comparison of feedforward and decoding weights. (ii) After learning, the decoding weights are more structured and broader than the SRFs in A(ii), compatible to the decoding filter of speech measured in auditory cortex [38]. Same sorting of neurons as in A(ii). (iii) After re-training to the new stimulus, a small number of decoding filter (neurons) "specialize" to the new stimulus, while the decoding weights of the others change only mildly. The network thereby minimizes its firing rate response to the new stimulus, while still providing an accurate representation of it.

effort to cover the "hole" made in the representation (Fig 8B). This compensation is a result of instantaneous des-inhibition in the lesioned network, not plasticity [28]. While this re-balancing limits coding errors, it still leads to an inefficient representation due to the large firing rates required from compensating neurons. In a second step, the recurrent learning rules kick in, and the network adapts its recurrent connections so that each neuron is again balanced on a spike-by-spike time scale. In a third step, the network also re-learns the feedforward weights through the slower feedforward learning rules. As a consequence, the final, adapted network again covers the input space uniformly, just with fewer neurons (Fig 8C).

Importantly, and as already shown in Fig 2, plasticity of the recurrent EI loop (including E to I connections) is more important for this process than plasticity on the feedforward weights.

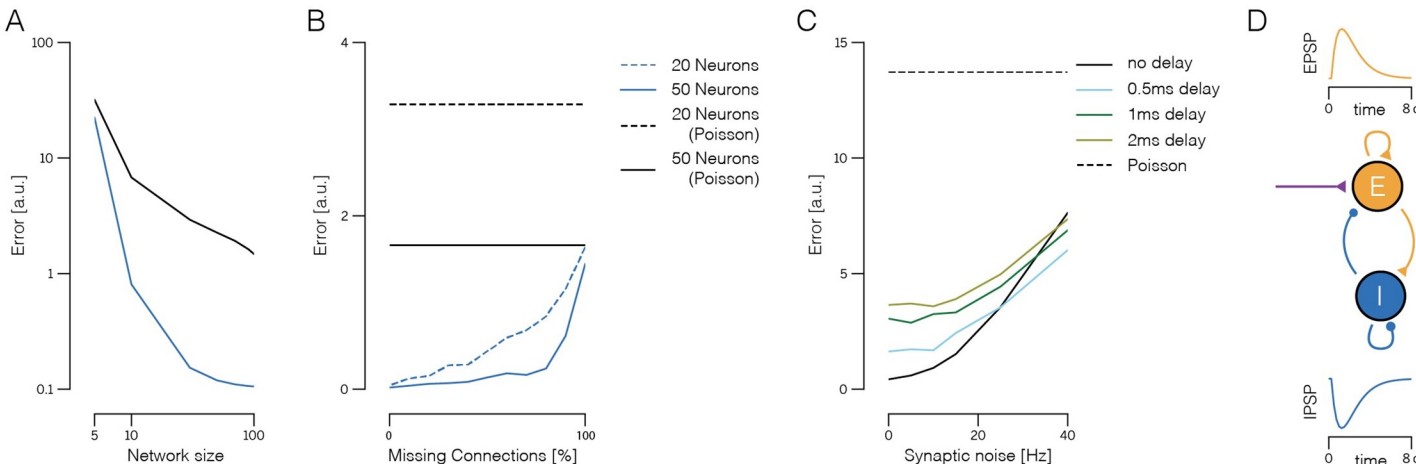

**Fig 7. Robustness of the learning rules to missing connections, noise, and synaptic delays.** All simulations are based on EI networks receiving two-dimensional, random input signals. Network size is given as number of inhibitory neurons. The pool of excitatory neurons is twice as large in all cases. **A.** Performance (mean-square error between input signal and signal estimate) of the learnt network as a function of (inhibitory) network size. Trained network (blue) and equivalent Poisson rate network (black), given by neurons whose firing follows Poisson processes with identical average rates. **B.** Performance of the learnt network as a function of connection sparsity. Here, we randomly deleted some percentage of the connections in the network, and then trained the remaining connections with the same learning rule as before. We adjusted the variance of the input signals to achieve the same mean firing rate in each neuron ($r = 5$ Hz in excitatory, $r = 10$ Hz in inhibitory neurons). Black lines denote the performance of an equivalent (and unconnected) population of Poisson-spiking neurons. **C.** Network performance as a function of synaptic noise and synaptic delay. Here, we injected random white-noise currents into each neuron. The size of the noise was defined as the standard deviation of the injected currents, divided by the time constant and firing threshold. Roughly, this measure corresponds to the firing rate cause by the synaptic noise alone, in the absence of connections or input signals. As in B, the input variance was scaled to get the same mean firing rate in each neuron ($r = 5$ Hz in excitatory, $r = 10$ Hz in inhibitory neurons). Different colors show curves for different synaptic delays (see panel D). **D.** Temporal profile of EPSCs and IPSCs (injected currents each time a spike is received) in the delayed networks, plotted as a function of the synaptic delay $d$. We rescaled the time axis to get the different delays used in panel C.

This observation leads to the following prediction: even in the absence of feedforward plasticity, the network recovers most of its efficiency (Fig 8D). While the tuning curves never achieve the perfect re-arrangement of the network with intact plasticity (Fig 8C), the responses of overactive cells are suppressed and shifted further towards the impaired direction. In contrast, if we were to block recurrent plasticity (Fig 8E), the network would become unbalanced and thereby inefficient due to the remaining action of the feedforward weights. While selectively blocking plasticity mechanisms at different synapses may seem a bit outlandish at first, the modern molecular biology toolbox does put it within reach [40]. By combining such techniques with focal lesions and awake recordings (e.g. calcium imaging) in local neural populations, these predictions are now within the range of the testable.

A second important prediction arises from the differential time course of feedforward and recurrent plasticity. Since recurrent plasticity should be much faster than feedforward plasticity, we predict that a partial recovery of the network efficiency will occur relatively fast (in minutes to hours of exposure to the stimuli with orientations matching the knocked-out cells). This will be performed mainly through a re-equalization of the population responses, but without major changes in the preferred tuning of the cells (compare Fig 8B and 8D). It will eventually be followed by a slower recovery of the tuning curve shapes and uniform density (but at a much slower time scale, e.g. over days or weeks of exposure).

## Discussion

In summary, we have shown how populations of excitatory and inhibitory neurons can learn to efficiently represent a set of signals spike by spike. We have measured efficiency with an objective function that combines the mean-square reconstruction error with various cost

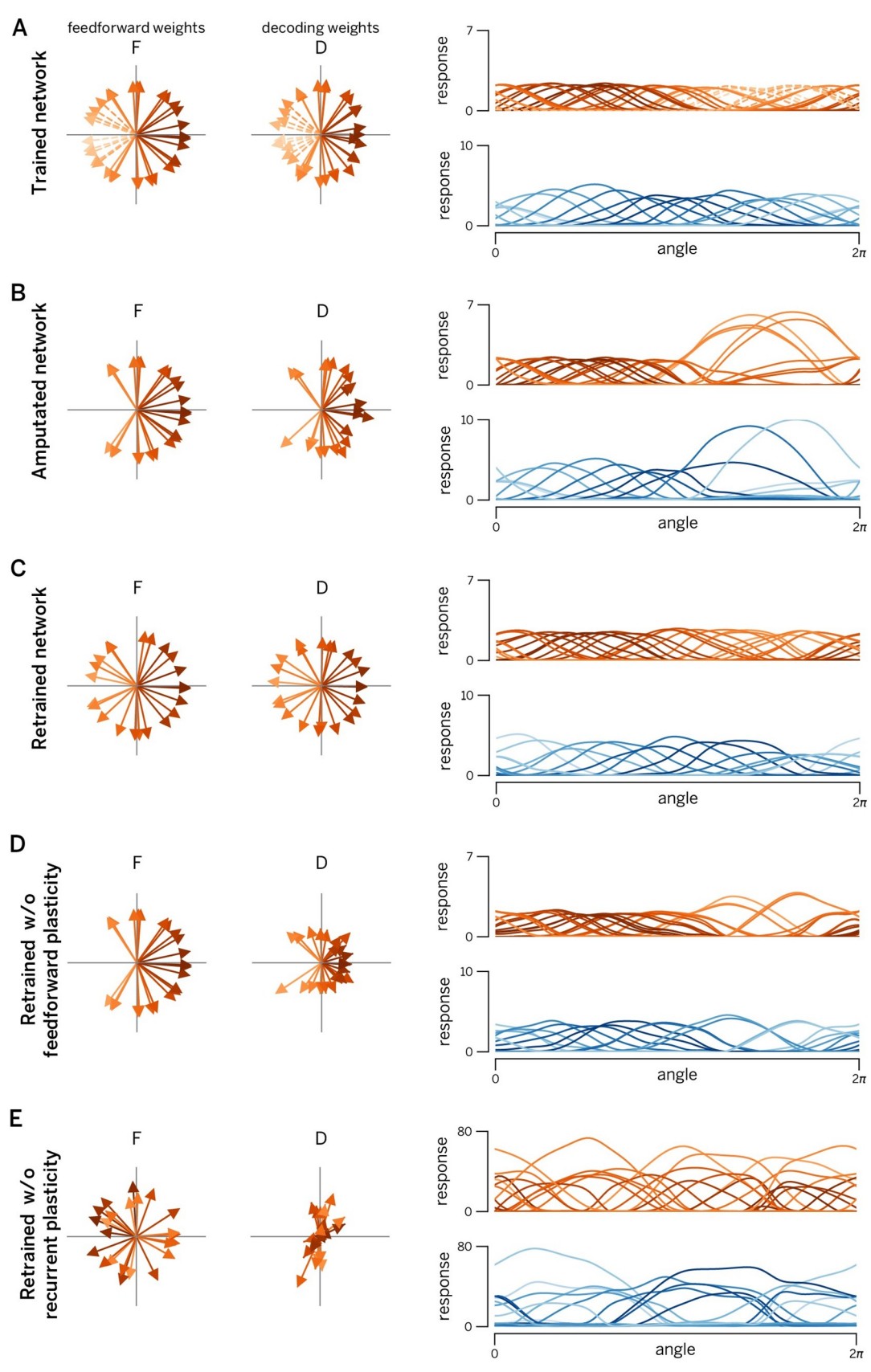

**Fig 8. Manipulating recurrent and feedforward plasticity. A.** EI network with 80 excitatory neurons and 40 inhibitory neurons, trained with two uncorrelated time-varying inputs. Left panels: learnt feedforward weights of excitatory population. Central panels: Optimal decoding weights of excitatory population. Right panels: Tuning curves of excitatory neurons (red) and inhibitory neurons (blue). Neurons encoding/decodings weights and tuning curves are shaded according to their preferred direction (direction of their decoding weight vector) in 2D input space. The color code is maintained in all subsequent figures (even if their preferred direction changed after lesion and/or retraining). **B.** Same network after deletion of leftward-coding excitatory neurons (see dashed lines in panel A). Note that no new training of the weights has yet taken place. Changes in tuning curves and decoding weights are due to internal network dynamics. We observe a large increase in firing rates and a widening and shifting of tuning curves towards the lesion, a signature that the network can still encode leftward moving stimuli, but does it in an inefficient way. **C.** Retrained network. The lesioned network in (B) was subjected to 1000s re-training of the connection. Consequently, the "hole" induced by the lesion has been filled by the new feedforward/decoding weights, all tuning curves once again covering the input space uniformly. **D.** Network with retrained recurrent connections (feedforward weights are the same as in panel B). Even without feedforward plasticity, the lesioned network is able to recover its efficiency to a large extent. **E.** Network with retrained feedforward weights only (recurrent connections are the same as in panel B). While feedforward weights once again cover the input space, absence of recurrent plasticity results in a massive increase in firing rates (and a concomitant decrease in coding precision). Consequently, training only feedforward weights after a lesion actually worsens the representation.

terms. While mathematically simpler than mutual-information-based approaches, our objective function includes both principal and overcomplete independent component analysis as special cases [41,42]. This type of unsupervised learning has previously been studied extensively in rate networks [10,16,17,43–48]. Implementations that seek to mimic biology by assuming spiking neurons, recurrent network architectures, and local learning rules, have always faced difficulties, and have therefore been largely limited to heuristic or approximative approaches, [12,13,15,36,49]. Using a rigorous, spike-based, and top-down approach, we have here derived biologically plausible learning rules that are guaranteed to converge to a specific connectivity and achieve a maximally efficient spike code. Importantly, single spikes are not to be considered as random samples from a rate, but are rather an integral part of a metabolically efficient brain.

We have limited our study on learning here to the encoding of time-varying signals into spikes. Several questions seem natural at this point. First, beyond peripheral sensory systems, most neurons receive spikes as inputs, not analog signals, which seems to violate one of our core premises. Second, neural systems perform computations with the signals they receive, rather than encoding them into spikes, only to be read out again at a later stage, which may seem a rather pointless exercise. Third, our learning rules have been derived in an unsupervised scenario, and one may wonder whether the core ideas underlying these rules can be extended beyond that.

Concerning spiking inputs, we note that nothing prevents us from replacing the analog input signals with spike trains. While we have chosen to explain these learning rules using analog inputs, our derivations were not dependent on this restriction. In fact, we have already used spike trains (rather than analog input signals) in the simulation of the EI-network in Fig 4—here the inhibitory neurons received spike trains from the excitatory neurons as 'feedforward' inputs, and we applied exactly the same feedforward learning rules as for the continuous-valued input signals (see also S1 Text, Section 13).

Concerning computations, we note that the solution to the encoding problem provides a necessary starting point for introducing more complex computations. For example, we showed previously that a second set of slower connections can implement arbitrary linear dynamics in optimally designed networks [27]. Non-linear computations can be introduced as well, but require that these non-linearities are implemented in synapses or dendrites [50]. The separation between coding and computation in these approaches is very similar to the separation used in the neural engineering framework [7].

Concerning learning, we note that there has been quite a lot of progress in recent years in developing local learning rules in supervised scenarios, both in feedforward [19–21] and recurrent networks [51,52]. We believe that the framework presented here provides crucial intuitions for supervised learning in spiking networks, since it shows how to represent global errors in local quantities such as voltages. In the future, these ideas may be combined with explicit single-neuron models [53] to turn local learning rules into global functions [21,52,54].

Apart from the theoretical advances in studying learning in spiking networks, many of the critical features that are hallmarks of cortical dynamics follow naturally from our framework, even though they were not included in the original objective. We list four of the most important features. First, the predicted spike trains are highly irregular and variable, which has indeed been widely reported in cortical neurons [4,55]. Importantly, this variability is a signature of the network's coding efficiency, rather than detrimental [32] or purposeful noise [56,57]. Second, despite this spike train variability, the membrane potentials of similarly tuned neurons are strongly correlated (due to shared inputs), as has indeed been found in various sensory areas [58,59]. Third, local and recurrent inhibition in our network serves to balance the excitatory feedforward inputs on a very fast time scale. Such EI balance, in which inhibitory currents track excitatory currents on a millisecond time scale, has been found in various systems and under various conditions [60,61]. Fourth, we have derived learning rules whose polarity depends on the relative timing of pre-and postsynaptic spikes (see insets in Fig 4A). In fact, the respective sign switches simply reflect the immediate sign reversal of the coding error (and thus of the membrane potential) after each new spike. As a result, even though our proposed learning rules are not defined in terms of relative timing of pre- and postsynaptic spikes, most connections display some features of the classic STDP rules, e.g., LTP for pre-post pairing, and LTD for post-pre pairing [62,63]. The only exception are E-E connections that exhibit "reverse STDP", i.e. potentiation for post-pre pairing (Fig 4A). Despite their simplicity, these rules are not only spike-time dependent but also weight and voltage-dependent, as observed experimentally [36].

Our framework thereby bridges from the essential biophysical quantities, such as the membrane voltages of the neurons, to the resulting population code, while providing crucial new insights on learning and coding in spiking neural networks.

## Materials and methods

Detailed mathematical derivations of the learning rules are explained in the supplementary materials (S1 Text). In addition, MATLAB code for the key simulations of the article is available on https://github.com/machenslab/spikes.

## Supporting information

**S1 Text. The supplementary material contains detailed mathematical derivations and proofs of all the main concepts explained in this article.** While it is referenced section by section, it can also be read as a separate, more technical explanation of the learning rules. (PDF)

## Acknowledgments

We thank Alfonso Renart, Bassam Attalah, Larry Abbott, and Nuno Calaim for comments on an earlier version of this manuscript.

## Author Contributions

**Conceptualization:** Wieland Brendel, Ralph Bourdoukan, Pietro Vertechi, Christian K. Machens, Sophie Denève.

**Data curation:** Ralph Bourdoukan, Christian K. Machens.

**Formal analysis:** Wieland Brendel, Pietro Vertechi.

**Funding acquisition:** Christian K. Machens, Sophie Denève.

**Investigation:** Wieland Brendel, Ralph Bourdoukan, Pietro Vertechi, Christian K. Machens, Sophie Denève.

**Methodology:** Wieland Brendel, Pietro Vertechi, Christian K. Machens, Sophie Denève.

**Project administration:** Christian K. Machens, Sophie Denève.

**Resources:** Christian K. Machens, Sophie Denève.

**Software:** Wieland Brendel, Ralph Bourdoukan, Christian K. Machens, Sophie Denève.

**Supervision:** Christian K. Machens, Sophie Denève.

**Validation:** Christian K. Machens, Sophie Denève.

**Visualization:** Wieland Brendel, Ralph Bourdoukan, Christian K. Machens, Sophie Denève.

**Writing – original draft:** Wieland Brendel, Ralph Bourdoukan, Pietro Vertechi, Christian K. Machens, Sophie Denève.

**Writing – review & editing:** Wieland Brendel, Pietro Vertechi, Christian K. Machens, Sophie Denève.

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
