## [Decision Letter · Decision Letter 0]

5 Oct 2019

Dear Dr Machens,

Thank you very much for submitting your manuscript 'Learning to represent signals spike by spike' for review by PLOS Computational Biology. Your manuscript has been fully evaluated by the PLOS Computational Biology editorial team and in this case also by independent peer reviewers. The reviewers appreciated the attention to an important problem, but raised some substantial concerns about the manuscript as it currently stands. While your manuscript cannot be accepted in its present form, we are willing to consider a revised version in which the issues raised by the reviewers have been adequately addressed. We cannot, of course, promise publication at that time.

Sincerely,

Samuel J. Gershman

Deputy Editor

PLOS Computational Biology

[LINK]

Reviewer's Responses to Questions

**Comments to the Authors:**

Reviewer #1: ‘Learning to represent signals spike by spike’ is a normative study on learning rules built to represent multiple signals simultaneously in a spiking neural network. This work starts where previous efforts from the same authors (particularly Boerlin et al. (2013)) had left off. In that previous study, it was shown that a carefully crafted arrangement of synaptic weights allows a network of spiking neurons to represent an arbitrary number of continuous time-dependent signals. These results relied on a precise arrangement of synaptic weights, and the authors had to assume that such an arrangement was given a priori. In the present study, they ask if there exists spike-timing dependent synaptic learning rules to let the network to self-organize to this rather convenient state. Following a normative approached based on a greedy optimization of decoding error, they show that there is a learning rule which can maximize encoding precision and is shows at the same time a voltage and spike-timing dependence in a way that matches, qualitatively, some standard in vitro experiments. The authors reports considerable achievements of their learning rule in spiking neural network (I commend the efforts to establish a Dalean network that self-organizes in precise input representation).

This work comes in an opportune moment as part of the field of computational neuroscience shows a growing interest in learning rules that will ensure that a particular function is conserved. Learning rules have been shown to have a plethora of shapes and properties, and the recent introduction of inhibitory learning rules is only making things worse. There was much focus on rate based learning rules (FORCE learning and its variants), which has recently been shown to work with spiking neurons. In the same vein, there is much recent research on learning weight matrices that are transpose of a known weight matrix in biological implementation of deep learning. All these problems are connected to the present work. Yet the present study is original and distinct from other studies in the sense that it applies to the predictive coding framework and promises of an energy efficient encoding of information.

That being said, I think the paper needs to be revisited carefully in order to unify the narrative, the results presented and and the supplementary material. I expand on my point of view below, but overall I recommend further consideration of this MS for PLoS CB.

1. Abstract. The abstract seems to confuse premises with results. Statements like ‘here we show that many single-neuron quantities including voltages… acquire a precise functional meaning’ summarizes the premise of the work rather than the result. Premise in the sense that these are the assumptions from which the main results are derived, but in that case also because these are the results of a previous paper from the same group. Similarly for the conclusion sentence of the abstract. Going a little further, the question of finding THE level at which THE functional meaning emerges is not a key question in neuroscience. There are multiple levels of description and therefore functionality has multiple levels of description. Multiple levels of description, but also multiple types of systems with membrane potentials (the spike-based predictive coding framework does not apply to non-spiking retina despite the shared coding and energy constraints). These statements is made even more out of place when we consider the fact that I don’t think the work presented in the MS addresses this question. The work is about whether biological-looking learning rules can give rise to the nice benefits of the predictive coding framework. In effect, it would be nice if the abstract would be more to the point. The introduction is good, so just a condensed version of the intro would do. Similar issue with the end of the discussion.

2. Intro. I thought I would mention a few related works that I think are germane to the present study:

- Membrane potential as prediction error: Urbanczik and Senn, Neuron (2014).

- Learning the transpose of weights: Burbank (2015) uses an STDP setup to do so.

- Further learning of the transpose of the weights in rate models; Akrout et al. (2019); Lansdell, Prakash and Kording (2019)

3. Decoder weights. In many places the decoder weights are said to be unknown, but then they are the target of the recurrent weights, later they are the target of feedforward weights. How can the weights be targets without being known? Similarly, the decoder network is sometimes an explicit network elements, but recurrently it is just a virtual presence introduced for the sake of argument. It was particularly confusing in the supplementary materials: D is assumed unknown but should follow S.23, which is in effect F. Then F is assumed unknown, but derived to be D. I am left with the impression that there is a circular argument in the learning of F with D that is not fixed a priori. In my point of view, the circular argument is present in sections like 8.2 of supplementals. Same with section 6, which (6.4 has F to mimic D, but 6.5 chooses D with F).

4. Supplementary material. I could not fully follow the supplementary material. There was too much back and forth between different formalisms and different sets of assumptions. Current based learning, then voltage-based learning, then L1-L2 costs, then summary of some of it, there has to be a more streamlined version. There are a few, perhaps interesting, theoretical results that are not part of the results as far as I can see. Particularly parts of section 5.

5. Figure 2 does not give enough credit to learning recurrent connections. As we can see in Fig. 2A, the error goes down dramatically at then end of the recurrence-learning, so the signal reproduction is near perfect before learning the FF weights. Since it is not clear in Figure 3 whether the recurrent weights have been adequately learned, this brings me to the question eluded to earlier: can you prove that FF learning is essential? In which case?

6. Figure 4 shows inverted I-E connections with respect to EE, but the text mentions the learning rules are the same. Also, on which side is t_post>t_pre in the learning windows?

7. x is assumed to white at many places, but it is simulated as a non white signal (section 14.1 and eq. S.1)

8. Having filtered x and non-filtered x denoted by the same variables in the main results section is disturbing, please fix.

9. Please explain why noise is included in the simulations. I presume it is required to some extent.

10. Please verify the transpose on Eq. S.12 and S.10.

11. L* not defined in first equation of Section 4 of supplementary materials. Wording is a bit confusing just before that equation as it is as if a loss function is defined by the equation. I find this equation and the one in 5.1 confusing. The goal is not to determine L*, but to determine D or o that achieves L*. Why not use argmin?

12. What is a population spike in (section S7)

13. Main results section p14-15. The term decoding weights has been used instead of FF weights.

14. Note that FF weights from thalamus are fixed after critical period.

15. I was a bit frazzled by the overly simple descriptions

Reviewer #2: Comments are uploaded as an attachment.

**Have all data underlying the figures and results presented in the manuscript been provided?**

Reviewer #1: Yes

Reviewer #2: None

PLOS authors have the option to publish the peer review history of their article (what does this mean?). If published, this will include your full peer review and any attached files.

Reviewer #1: Yes: Richard Naud

Reviewer #2: No

---

## [Editor Report · Decision Letter 1]

27 Jan 2020

Dear Dr Machens,

We are pleased to inform you that your manuscript 'Learning to represent signals spike by spike' has been provisionally accepted for publication in PLOS Computational Biology.

Before your manuscript can be formally accepted you will need to complete some formatting changes, which you will receive in a follow up email. A member of our team will be in touch within two working days with a set of requests.

Best regards,

Samuel J. Gershman

Deputy Editor

PLOS Computational Biology

---

## [Editor Report · Acceptance letter]

9 Mar 2020

PCOMPBIOL-D-19-01208R1 

Learning to represent signals spike by spike

Dear Dr Machens,

I am pleased to inform you that your manuscript has been formally accepted for publication in PLOS Computational Biology. Your manuscript is now with our production department and you will be notified of the publication date in due course.

With kind regards,

Laura Mallard
